# Ethyl Acetate Fraction of *Amomum xanthioides* Ameliorates Nonalcoholic Fatty Liver Disease in a High-Fat Diet Mouse Model

**DOI:** 10.3390/nu12082433

**Published:** 2020-08-13

**Authors:** Hwi-Jin Im, Seung-Ju Hwang, Jin-Seok Lee, Sung-Bae Lee, Ji-Yun Kang, Chang-Gue Son

**Affiliations:** Liver and Immunology Research Center, Dunsan Oriental Hospital of Daejeon University Daedukdae-ro 176 bun-gil 75, Seo-gu, Daejeon 35353, Korea; lastdohee@gmail.com (H.-J.I.); bluesea9292@naver.com (S.-J.H.); neptune@dju.kr (J.-S.L.); sky161300@naver.com (S.-B.L.); kangjy0118@naver.com (J.-Y.K.)

**Keywords:** NAFLD, NASH, fatty liver, herb, *Amomum xanthioides*

## Abstract

The global prevalence of nonalcoholic fatty liver disease (NAFLD) is estimated to be 25% and has continued to increase; however, no drugs have yet been approved for NAFLD treatments. The ethyl acetate fraction of *Amomum xanthioides* (EFAX) was previously reported to have an anti-hepatic fibrosis effect, but its effects on steatosis or steatohepatitis remain unclear. This study investigated the anti-fatty liver of EFAX using a high-fat diet mouse model. High-fat diet intake for 8 weeks induced hepatic steatosis with mild inflammation and oxidative damage and increased the adipose tissue weight along with the development of dyslipidemia. EFAX treatment significantly ameliorated the steatohepatic changes, the increased weight of adipose tissues, and the altered serum lipid profiles. These observed effects were possibly due to the lipolysis-dominant activity of EFAX on multiple hepatic proteins including sterol regulatory element-binding protein (mSREBP)-1c, peroxisome proliferator-activated receptor (PPAR)-α, AMP-activated protein kinase, and diglyceride acyltransferases (DGATs). Taken together, these results show that EFAX might be a potential therapeutic agent for regulating a wide spectrum of NAFLDs from steatosis to fibrosis via multiple actions on lipid metabolism-related proteins. Further studies investigating clear mechanisms and their active compounds are needed.

## 1. Introduction

Nonalcoholic fatty liver disease (NAFLD) is becoming a major health problem because of its high estimated global prevalence of 25% and continuous increasing pattern over the last decade [1]. Recently, the economic burden of NAFLD has been reported to be over 100 billion dollars in the USA [2]. Furthermore, NAFLD is crucially linked to the development of not only chronic hepatic diseases but also obesity, type 2 diabetes, cardiovascular diseases, and chronic kidney diseases [3,4,5].

Weight loss through lifestyle modification (dietary habits and physical exercise) is the first-line treatment for NAFLD [6]. To improve fibrosis and steatosis, weight loss of 10% or more is required [7]. However, a previous study reported that only 10% of patients achieved a weight reduction of more than 10%, and not all patients gained beneficial effects from the weight reduction [8]. In addition, 8~19% of nonobese subjects, especially in Asia, are also diagnosed with NAFLD [9]. Therefore, although weight loss through lifestyle modification is a very effective therapy for NAFLD, pharmacological intervention in addition to lifestyle modification is also recommended for patients with NAFLD [10].

Clinically, vitamin E and pioglitazone have been recommended for patients with NAFLD by American and European guidelines [11,12]. The above two drugs were reported to show beneficial effects on nonalcoholic fatty liver (NAFL) and nonalcoholic steatohepatitis (NASH). However, there is insufficient evidence on their modulation of fibrosis or cirrhosis [13], and several adverse effects including increased mortality or cancer risk have been reported [14,15,16]. Therefore, the development of other treatments is still required.

A total of 395 clinical trials were registered at ClinicalTrials.gov (at 10 June 2020) for the investigation of a potential drug for NAFLD. However, to date, no agent has been approved for the treatment of NAFLD. A major challenge for NAFLD drug development is the complexity and multitude of therapeutic targets; therefore, combination therapy or herbal medicine is considered a promising therapy for NAFLD [17,18,19].

*Amomum xanthioides* is a medicinal herb that has been reported to show gastrointestinal protection, liver protective, and anti-dyslipidemic effects [20]. Moreover, our previous studies proved that one *A. xanthioides* fraction has anti-hepatofibrotic activity in dimethylnitrosamine (DMN) and bile duct-ligation animal models [21,22]. Hepatic fibrosis is considered an advanced pathologic stage of NAFLD [13]. We thus hypothesized that *A. xanthioides* is able to control NAFL and NASH.

We herein investigated the anti-NAFLD effects of *A. xanthioides* using a high-fat diet mouse model.

## 2. Materials and Methods

### 2.1. Preparation and Fingerprinting of A. xanthioides

*A. xanthioides* was purchased from Jeong-Seong Pharmaceutical Company (Daejeon, Korea). After screening the optimized fractions, the ethyl acetate fraction of *A. xanthioides* (EFAX) was selected and extracted using an organic solvent extraction method (Appendix A). Briefly, 10 kg of ground *A. xanthioides* was extracted in 100 L of absolute methanol for 1 week. One hundred milliliters of distilled water (DW) was mixed with 900 mL of the methanol fraction. Then, 100 mL of a petroleum ether extract was mixed with 900 mL of DW (1 L) and further fractionated with ethyl acetate (1 L) to isolate EFAX. EFAX was stored at −70 °C in the Liver & Immunology Research Center, Daejeon Oriental Hospital of Daejeon University (Storage specimen #EFAX 2017-03, Liver and Immunology Research Center, Daejeon, Korea). The final fraction yields were 0.19% (*w*/*w*).

To confirm the reproducibility of the EFAX chemical composition, ultrahigh-performance liquid chromatography–tandem mass spectrometry (UHPLC-MS/MS, Thermo Scientific, Waltham, CA, USA) was performed as described previously [21]. Briefly, quantitative analysis of the major three compounds in EFAX was performed using UHPLC-MS/MS. The chromatogram indicated that the main chemical components of EFAX were procyanidin B2, catechin, and quercitrin and occurred at 4.31, 4.75, and 7.24 min, respectively. The molecular weights of the three chemicals were confirmed as follows: procyanidin B2, 578.17 g/mol; catechin, 290.08 g/mol; and quercitrin, 448.11 g/mol (Appendix A).

### 2.2. Chemicals and Reagents

The reagents for the present study were as follows: TRI reagent (Invitrogen, Carlsbad, CA, USA); RIPA buffer (LPS solution, Korea); bovine serum albumin (BSA, GenDEPOT, Katy, TX, USA); n-butanol (J.T. Baker, Phillpsburg, NJ, USA); Mayer’s hematoxylin, methanol, and isopropanol (Wako Pure Chemical Industries, Osaka, Japan); and phosphoric acid and potassium chloride (SAMCHUN, Seoul, Korea) All other materials including Oil Red O powder, hematoxylin, sulfanilamide, N-ethylenediamine dihydrochloride, sodium nitrite, (±)-α-tocopherol (vitamin E), catechin, quercitrin, and procyanidin B2 were purchased from Sigma-Aldrich (St. Louis, MO, USA).

### 2.3. Animals and Experimental Design

A total of forty-eight C57BL/6N male mice (6-week old; 22–24 g) were purchased from Daehanbio-link (Eumseong-gun, Chung-book, Korea). The mice had free access to food pellets (Daehanbio-link) and water and were housed in a room maintained at 22 ± 2 °C under a 12 h light: 12 h dark cycle. After acclimatization for 1 week, mice were divided randomly into six groups (*n* = 8): the naive, control, EFAX (25, 50, and 100 mg/kg), and vitamin E (100 mg/kg) groups. Mice were fed a 60% high fat and 20% low carbohydrate diet (Research Diets Inc., New Brunswick, NJ, USA) for 8 weeks, excluding the naive group (normal chow diet, Appendix A). From the 3rd week of high fat diet feeding, distilled water (control), EFAX, and vitamin E were administered by gavage daily for the last 6 weeks. Food intake and body weight were measured twice a week. On the final day of the experiment, the mice were euthanized in CO_2_ Chamber (Jeungdo Bio&Plant, Seoul, Korea) after a 6-h fast, and whole blood was collected from the abdominal aorta. The liver and adipose tissues (epididymal, retroperitoneal and visceral) were weighed and stored immediately.

The protocol was approved by the Institutional Animal Care and Use Committee of Daejeon University (DJUARB2020-008) and was conducted in accordance with the Guide for the Care and Use of Laboratory Animals, published by the National Institutes of Health (NIH, MD).

### 2.4. Determination of Hepatic Lipid Levels and Serum Biochemistry

Hepatic triglyceride (TG) and hepatic total cholesterol (TC) levels were determined using commercial kits (ASAN pharmacy, Seoul, Korea) according to a previous method [23]. Serum levels of aspartate aminotransferase (AST), alanine aminotransferase (ALT), TG, TC, low-density lipoprotein cholesterol (LDL), and high-density lipoprotein cholesterol (HDL) were determined using an autoanalyzer (Chiron, Emeryville, CA, USA).

### 2.5. Histopathological Analysis in the Hepatic Tissue

To evaluate hepatic fat accumulation, frozen liver samples were sectioned at 10 μm and stained with Oil-Red O and Mayer’s hematoxylin. For the histomorphological evaluation, the slides were stained with hematoxylin and eosin (H&E). For immunohistochemical staining of myeloperoxidase (MPO) and 4-hydroxynonenal (4-HNE), the liver tissues were fixed in 10% formalin, embedded in paraffin blocks, and cut into 5 μm sections. Immunostaining was performed with anti-4-HNE antibody (Abcam, Cambridge, UK) or anti-MPO antibody (Abcam, Cambridge, UK), and a biotinylated secondary antibody (Vector Laboratories, Burlingame, CA, USA) was used. The slides were developed with 0.05% DAB (Sigma Aldrich, St. Louis, MO, USA) and then immediately counter-stained with Mayer’s hematoxylin. Five photographs per sample were acquired using an optical microscope (Leica, Wetzlar, Germany) at 200× (for H&E and 4-HNE) or 400× (for Oil-Red O and MPO) magnification.

### 2.6. Determination of Inflammatory Cytokines in Hepatic Tissue

Hepatic levels of inflammatory cytokines were measured using commercial ELISA kits: tumor necrosis factor (TNF)-α (BD Biosciences, San Jose, CA, USA) and interleukin (IL)-1β (R&D Systems, Minneapolis, MN, USA).

### 2.7. Determination of Nitric Oxide (NO) and Lipid Peroxidation in Hepatic Tissue

The hepatic levels of NO were evaluated using the Griess method [24]. Hepatic levels of lipid peroxidation represented as malondialdehyde (MDA) were determined using thiobarbituric acid reactive substances (TBAR) according to a previous method [25]. The absorbance was measured at 540 nm for NO or 520‒535 nm for MDA using a spectrophotometer (Molecular Device Corp., San Jose, CA, USA).

### 2.8. Western Blot Analysis for Lipid Metabolic Proteins in the Hepatic Tissue

#### 2.8.1. Fatty Acid Metabolism-Related Proteins

Western blot analysis evaluated five hepatic proteins related to fatty acid metabolism as follows: mature sterol regulatory element-binding protein (mSREBP)-1c, fatty acid synthase (FAS), peroxisome proliferator-activated receptor (PPAR)-α, total AMP-activated protein kinase (AMPK)-α, and phosphorylated-AMPK (pAMPK)-α.

#### 2.8.2. TG Synthesis-Related Proteins

Regarding TG synthesis, western blot analysis was conducted to evaluate three hepatic proteins as follows: glycerol-3-phosphate acyltransferase (GPAM), diglyceride acyltransferase (DGAT)1, and DGAT2.

#### 2.8.3. Western Blot Performance

The proteins were separated by 10% polyacrylamide gel electrophoresis and transferred to polyvinylidene fluoride (PVDF) membranes. After blocking in 3% BSA, the membranes were probed with primary antibodies (SREBP-1c, PPAR-α, GPAM, DGAT1, and DGAT2 (Abcam, UK); FAS, AMPK-α, and pAMPK-α (Cell Signaling Technology, Danvers, MA, CA, USA); β-actin from (Thermo Fisher Scientific, Waltham, MA, USA)) overnight at 4 °C. The membranes were washed and incubated with an anti-rabbit IgG-HRP (GeneTex, Irvine, CA, USA) for 2 h. Proteins were imaged using an advanced enhanced chemiluminescence (ECL) advanced kit (Thermo Fisher Scientific, CA, USA). Protein expression was observed using the FUSION Solo System (Vilber Lourmat, Marne-la-Vallée, France), and semi-quantified using ImageJ (NIH).

### 2.9. Statistical Analysis

The data are expressed as the means ± standard deviation (SD) or fold changes in means. Statistical significance was determined by using one-way analysis of variance (ANOVA) followed by *Dunnett’s* test. In all analyses, *p* < 0.05 was taken to indicate statistical significance.

## 3. Results

### 3.1. Food Intake and Body, Liver, and Adipose Tissue Weights

As expected, the 8-week high-fat diet increased the average body weight to 1.5-fold that of the naive group fed the normal diet, and the weights of liver and adipose tissues (epididymal, retroperitoneal, visceral, and total, *p* < 0.05 or *p* < 0.01) also increased. Administration of EFAX (especially 25 mg/kg and 100 mg/kg) significantly attenuated these increases in all measurements: body weight (*p* < 0.05), absolute liver weight (*p* < 0.05 or *p* < 0.01), and adipose tissue weight (*p* < 0.05), respectively (Figure 1A,B and Table 1). The positive control group receiving vitamin E (100 mg/kg) did not show any significant differences for those measurements.

In this experimental model, the consumed food weight differed between the naive (normal diet) and high-fat diet groups, but there was no significant difference between the control and treatment groups. Total calorie intake was not significantly different among all groups, including the naive group (Figure 1C,D). 

### 3.2. Effects on Hepatic Lipid Accumulation

The high fat diet intake for 8 weeks remarkably induced the hepatic lipid accumulation, as evidenced by histopathologic findings using Oil Red O staining, with high levels of hepatic TG (4.6-fold) and TC (7.7-fold) compared to those in the naive group. EFAX treatments significantly ameliorated the deposition of hepatic lipid droplets (*p* < 0.01) and TG levels (*p* < 0.01) compared with those of the control. An effect of EFAX on hepatic TC levels was observed but did not show statistical significance. Vitamin E showed no effects on Oil Red O staining and hepatic TG levels but significantly reduced hepatic TC levels (*p* < 0.01, Figure 2).

### 3.3. Effects on Lipid Profiles in Serum

The high-fat diet considerably elevated serum levels of TC (2.0-fold), LDL (2.8-fold), and HDL (1.4-fold) but reduced serum TG levels (0.6-fold). EFAX treatments significantly reduced these alterations in TC (*p* < 0.01) and LDL (*p* < 0.01) levels compared with those of the control, but no changes in either HDL or TG levels were observed. No significant effect on lipid profile was observed in the vitamin E-treated group (Table 1).

### 3.4. Effects on Molecules for Fatty Acid-Metabolism

The 8-week administration of a high-fat diet activated both “*de novo* lipogenesis”-related proteins (mSREBP-1c; 4.3-fold and FAS; 1.2-fold) and a “fatty acid β-oxidation”-related protein (PPAR-α; 2.4-fold). EFAX treatments markedly reduced the alterations of hepatic SREBP-1c (*p* < 0.01 for 50 and 100 mg/kg) and FAS (*p* < 0.01) but significantly upregulated the PPAR-α levels (*p* < 0.01) compared with those of the control (Figure 3A–C). EFAX treatments also accelerated AMPK-α, a molecule with a pro-lipolysis activity, evidenced by a marked elevation in the pAMPK/AMPK ratio (*p* < 0.01) compared with that of the control. Vitamin E also showed similar effects on mSREPB-1c (*p* < 0.05) and FAS (*p* < 0.01) as EFAX but reduced the activation of hepatic PPAR-α (*p* < 0.01). Vitamin E had no effect on the pAMPK/AMPK ratio (Figure 3A–C).

### 3.5. Effects on Molecules for TG Synthesis

Administration of a high-fat diet significantly elevated the hepatic protein levels of GPAM (3.6-fold, playing a role in TG synthesis before secretion or storage) and DGAT2 (1.9-fold, TG storage) but reduced that of the DGAT1 protein (0.8-fold, TG secretion). These alterations were significantly ameliorated by EFAX treatments (especially at 50 mg/kg or 100 mg/kg, *p* < 0.05 or *p* < 0.01) compared with the control. Vitamin E also showed significant effects on the hepatic levels of GPAM (*p* < 0.05) and DGAT2 (*p* < 0.01) but not on DGAT1 (Figure 3A,D).

### 3.6. Effects on Hepatic Inflammation

H&E staining reveals an 8-week high-fat diet induced mild hepatic inflammation (the number of ballooned hepatocytes was increased but no hepatic fibrosis was observed), which was consistent with elevated serum levels of AST (1.5-fold) and ALT (3.2-fold) in the control (Figure 4). This was supported by the measurements of hepatic MPO (10.5-fold), TNF-α (1.9-fold), and IL-1β (1.3-fold), respectively (Figure 5). EFAX treatments considerably ameliorated these changes in H&E staining; serum levels of AST (*p* < 0.05 or *p* < 0.01) and ALT (*p* < 0.05 or *p* < 0.01); and hepatic levels of MPO activities (*p* < 0.01), TNF-α (*p* < 0.05 or *p* < 0.01), and IL-1β (*p* < 0.05 or *p* < 0.01) compared with those of the control. No beneficial result, however, was observed in the vitamin E treatment group (Figure 4 and Figure 5).

### 3.7. Effects on Hepatic Oxidative Stress

As expected, an 8-week high-fat diet induced notable hepatic oxidative stress, evidenced by increases in hepatic levels of 4-HNE (12.7-fold), MDA (1.7-fold), and NO (2.1-fold). These alterations, namely, the levels of 4-HNE (*p* < 0.01), MDA (*p* < 0.05), and NO (*p* < 0.01), were significantly attenuated by the EFAX treatments compared with those of the control. Vitamin E exerted considerable effects on hepatic 4-HNE (*p* < 0.01) and MDA (*p* < 0.01) levels but not on NO levels (Figure 6).

## 4. Discussion

NAFLD is defined as hepatic disorders caused by fat accumulation (≥5% of the liver weight) in the absence of excessive alcohol consumption and its sequential diseases, including inflammation (NASH) and fibrotic changes (liver fibrosis and cirrhosis) [26,27]. The presence of NAFLD increases the risk of type 2 diabetes (1.6~5.5-fold) and mortality from cardiovascular diseases (approximately 2.0-fold) and chronic kidney diseases (1.4~1.9-fold) [4,28,29]. Clinical data reported that 20~80% of patients with NAFLD also have dyslipidemia [30]. As expected, the 8-week high-fat diet induced fatty liver and dyslipidemia in the present study. Then, EFAX treatments effectively protected the development of both fatty liver and the altered serum profiles of lipid contents except for TG (Figure 2). On the other hand, the present animal model significantly reduced serum TG levels, which can result from the lower amounts of carbohydrates (26.2% vs. 72.0%) in the high-fat diet than in the normal diet (Table 1, Appendix A). In fact, a number of clinical studies reported that a high-fat diet could lower serum TG levels [31,32,33].

NAFLD is a multisystem disease that involves dynamic interactions between the liver and various extrahepatic organs, including blood, adipose tissues, and muscles [34]. Obesity, excessive fat accumulation in adipose tissues, is an independent risk factor for NAFLD, and the global prevalence of obesity has increased up to 39.0% [35]. Current lifestyles, such as sedentary behavior and overnutrition, are considered the main contributors to excess fat storage in body organs [36]. Accordingly, weight loss is the most powerful intervention for NAFLD, and can be obtained by diet therapy and/or exercise [7]. In this study, EFAX considerably ameliorated the increases in body weight, adipose tissue weight, and liver weight associated with a high-fat diet (Figure 1A,B, Table 1). These outcomes, including weight loss, did not result from a reduction in total dietary caloric intake (Figure 1D), which suggests the potential of EFAX as a therapeutic drug for NAFLD. Several nature-derived compounds, such as gallic acid and paeoniflorin, were previously reported to also exert reducing effects on body weight and adipose tissue weight, as well as on fatty liver, in a high-fat diet model [37,38]; however, berberine, a popular nature-derived agent used to treat NAFLD, did not affect body weight or liver weight [39].

The liver is a central organ for homeostasis of lipid metabolism through both lipogenesis and lipolysis as well as through import from or export to blood and secretion with bile acid [40]. To explore the mechanisms corresponding to the above results by EFAX, we analyzed molecules for lipid metabolism in the liver. In our data, EFAX exerted lipolysis-dominant activity with inhibition of de novo lipogenesis, as evidenced by modulations of PPARα and AMPK versus mSREBP-1c and FAS (Figure 3). These molecules have been pharmaceutical targets, such as metformin targeting AMPK, a key modulator of fatty acid β-oxidation, for NAFLD [41]. In fact, AMPK activity (pAMPK/AMPK levels) was upregulated in our study, which would be the compensatory reaction against an overload of fat due to a high-fat diet, and this upregulation was further accelerated by EFAX treatments (Figure 3C). TG is the major lipid component in the liver of patients with NAFLD [42], and EFAX showed activity controlling hepatic lipid homeostasis (Figure 3). The molecules GPAM, DGAT2, and DGAT1 are well known to play important roles in the initiation of TG synthesis, storage of lipid droplets, and TG secretion into VLDL, respectively [43].

Broadly, NAFLD can be divided into simple steatosis (NAFL) and steatohepatitis (NASH), accounting for approximately 80~90% and 10~20% of all NAFLD cases [44]. NAFL itself is not medically problematic, but approximately 25% of NAFL develops into NASH and further progresses to fibrosis in 3 years, which are the critical issues in NAFLD [45]. This progression involves multiple contributors, such as oxidative stress and inflammation following the dysregulation of lipid metabolism [28]. The overload of free fatty acids in the liver can lead to inflammation through the induction of phagocytic activation and oxidative stress [46,47,48]. Although it is not certain whether inflammation is always preceded by fat accumulation, lipid-overload liver injury is still a dominant model explaining NASH pathogenesis [46]. As expected, EFAX exerted anti-inflammatory activities evidenced by hepatic enzymes of serum AST and ALT levels, along with inflammatory cytokines (TNF-α, IL-1β), MPO activity, and oxidative stress (4-HNE, MDA, NO), respectively (Figure 4, Figure 5 and Figure 6). Prolonged inflammation in the liver could progress into hepatic fibrosis, cirrhosis, and sometimes hepatocarcinoma which determine liver disorder-related mortality [49]. Our previous studies presented anti-hepatofibrotic effects of EFAX using bile-duct ligation and DMN liver injury animal models [21,22]; however, this study first reports the therapeutic effects of EFAX against NAFLD.

In our research regarding drug development for NAFLD treatment, we evaluated the potential of EFAX from *A. xanthioides,* a traditional medicinal herb treating gastrointestinal, hepatic, and dyslipidemic disorders [20]. The main constituents of EFAX include procyanidin B2, catechin, and quercitrin (Appendix A). Previous studies reported anti-fatty liver and anti-fibrotic effects of procyanidin B2 using animal models [50,51]. Several clinical studies revealed the beneficial effects of catechin-enriched green tea on NAFLD [52,53]. However, we still do not know the exact compounds accounting for the anti-NAFLD effects in EFAX, which we will investigate in the future. In the present study, we adapted vitamin E as a positive control. Vitamin E, a well-known anti-oxidant, has been clinically recommended as a plausible treatment for NAFLD [11]. In our experimental model, vitamin E showed notable effects on hepatic levels of TC and oxidative stress but no significant effects on hepatic levels of TG and inflammatory cytokines (TNF-α and IL-1β). A number of previous clinical studies have suggested anti-NAFLD effects of vitamin E; however, these effects are still controversial [54,55,56]. We cannot assure the reasons for the non-positive effects of vitamin E on hepatic TG levels in our study, but it was also controversial in other animal studies depending on study groups [57,58]. On the other hand, gut dysbiosis is attracting attention as a critical contributor to NAFLD [59], and a previous study revealed procyanidin B2 improve NAFLD via regulating gut microbiome [50]. Modulation of gut dysbiosis might be an underlying mechanism which could explain the anti-NAFLD effect of EFAX. Future study is needed to explore this possibility.

Taken together, the results of the present study revealed the therapeutic effects of EFAX on high-fat diet-induced NAFLD. The main mechanisms corresponding to these effects may involve the modulation of molecules related to hepatic lipolysis and lipogenesis as well as the secretion of lipids. Further studies are required to better understand the mechanisms and identifications of its active compounds.

## Figures and Tables

**Figure 1 nutrients-12-02433-f001:**
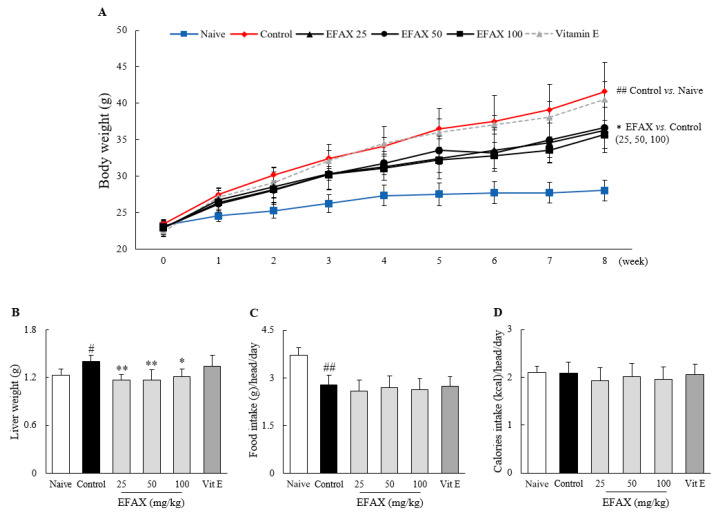
Effects on body weight, liver weight, and food (and calorie) intake. Mice (*n* = 8 in each group) were fed a 60% high-fat diet for 8 weeks, excluding the naive group (normal chow diet). From the 3rd week of high-fat diet feeding, distilled water (control), ethyl acetate fraction of *Amomum xanthioides* (EFAX) (25 mg/kg, 50 mg/kg, 100 mg/kg), and vitamin E (100 mg/kg) were administered by gavage daily for the last 6 weeks. Body weight (**A**), liver weight (**B**), and food intake (**C**) were recorded, and then caloric intake (**D**) was calculated. ^#^
*p* < 0.05, ^##^
*p* < 0.01 compared with the naive group; * *p* < 0.05, ** *p* < 0.01 compared with the control group.

**Figure 2 nutrients-12-02433-f002:**
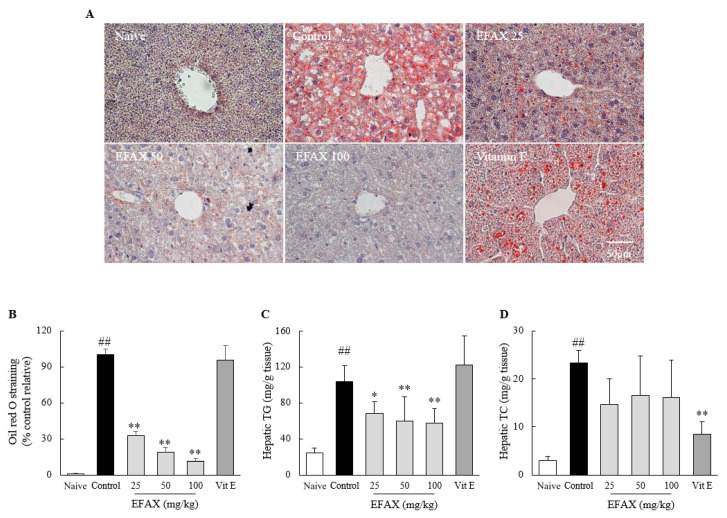
Effects on hepatic lipid profiles. Mice (*n* = 8 in each group) were fed a 60% high fat diet for 8 weeks excluding naive group (normal chow diet). From the 3rd week of high fat diet feeding, distilled water (control), EFAX (25 mg/kg, 50 mg/kg, 100 mg/kg), and vitamin E (100 mg/kg) were administered by gavage daily for the last 6 weeks. Oil Red O staining (**A**) of liver sections was performed, and representative photographs (400× magnification) were semi-quantified (**B**). Hepatic levels of Hepatic triglyceride (TG) (**C**) and total cholesterol (TC) (**D**) were evaluated. ^##^
*p* < 0.01 compared with the naive group; * *p* < 0.05, ** *p* < 0.01 compared with the control group.

**Figure 3 nutrients-12-02433-f003:**
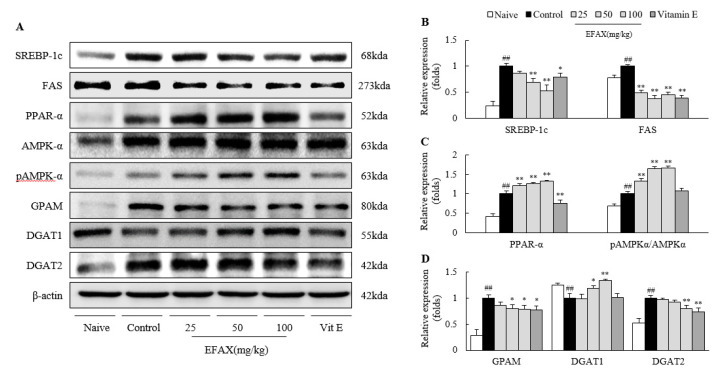
Effects on hepatic levels of lipid metabolic proteins. Mice (*n* = 8 in each group) were fed a 60% high-fat diet for 8 weeks, excluding the naive group (normal chow diet). From the 3rd week of high-fat diet feeding, distilled water (control), EFAX (25 mg/kg, 50 mg/kg, 100 mg/kg), and vitamin E (100 mg/kg) were administered by gavage daily for the last 6 weeks. Western blot analysis evaluated the hepatic level of eight-lipid metabolic proteins (**A**), and semi-quantifications were performed on sterol regulatory element-binding protein (mSREBP)-1c and fatty acid synthase (FAS) (**B**); peroxisome proliferator-activated receptor (PPAR)-α, AMP-activated protein kinase (AMPK)-α, and phosphorylated-AMPK (pAMPK)-α (**C**); and glycerol-3-phosphate acyltransferase (GPAM), diglyceride acyltransferase (DGAT)1, and DGAT2 (**D**). ^##^
*p* < 0.01 compared with the naive group; * *p* < 0.05, ** *p* < 0.01 compared with the control group.

**Figure 4 nutrients-12-02433-f004:**
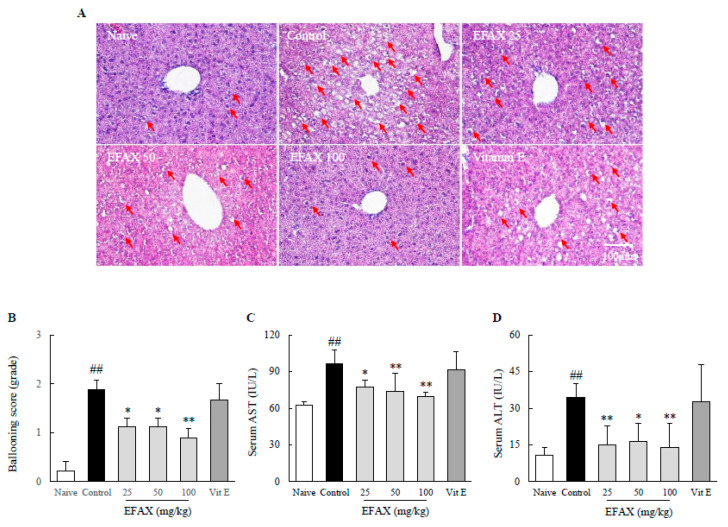
Effects on hepatic injuries. Mice (*n* = 8 in each group) were fed a 60% high-fat diet for 8 weeks, excluding naive group (normal chow diet). From the 3rd week of high-fat diet feeding, distilled water (control), EF AX (25 mg/kg, 50 mg/kg, 100 mg/kg), and vitamin E (100 mg/kg) were administered by gavage daily for the last 6 weeks. H&E staining (**A**) of liver sections was performed, and representative photographs (200× magnification) were semi-quantified (**B**). The red arrows indicate ballooned hepatocytes. Serum levels of aspartate aminotransferase (AST) (**C**) and alanine aminotransferase (ALT) (**D**) were evaluated. ^##^
*p* < 0.01 compared with the naive group; * *p* < 0.05, ** *p* < 0.01 compared with the control group.

**Figure 5 nutrients-12-02433-f005:**
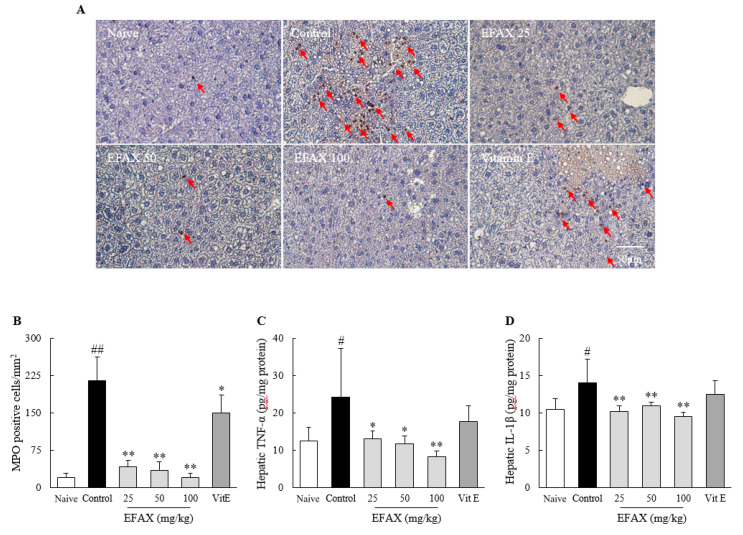
Effects on myeloperoxidase (MPO) staining and inflammatory cytokines. Mice (*n* = 8 in each group) were fed a 60% high-fat diet for 8 weeks, excluding the naive group (normal chow diet). From the 3rd week of high-fat diet feeding, distilled water (control), EFAX (25 mg/kg, 50 mg/kg, 100 mg/kg), and vitamin E (100 mg/kg) were administered by gavage daily for the last 6 weeks. MPO staining (**A**) of liver sections was performed, and representative photographs (400× magnification) were semi-quantified (**B**). Hepatic levels of tumor necrosis factor (TNF)-α (**C**) and interleukin (IL)-1β (**D**) were evaluated. ^#^
*p* < 0.05, ^##^
*p* < 0.01 compared with the naive group; * *p* < 0.05, ** *p* < 0.01 compared with the control group.

**Figure 6 nutrients-12-02433-f006:**
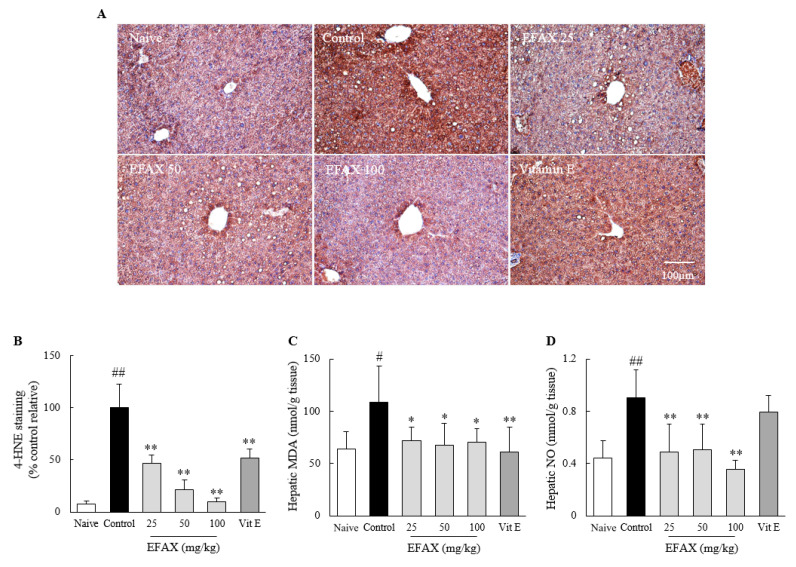
Effects on hepatic oxidative stress. Mice (*n* = 8 in each group) were fed a 60% high-fat diet for 8 weeks, excluding the naive group (normal chow diet). From the 3rd week of high-fat diet feeding, distilled water (control), EFAX (25 mg/kg, 50 mg/kg, 100 mg/kg), and vitamin E (100 mg/kg) were administered by gavage daily for the last 6 weeks. 4-hydroxynonenal (4-HNE) staining (**A**) of liver sections was performed, and representative photographs (200× magnification) were semi-quantified (**B**). Hepatic levels of malondialdehyde (MDA) (**C**) and Nitric Oxide (NO) (**D**) were evaluated. ^#^
*p* < 0.05, ^##^
*p* < 0.01 compared with the naive group; * *p* < 0.05, ** *p* < 0.01 compared with the control group.

**Table 1 nutrients-12-02433-t001:** Adipose tissue weights and serum lipid profiles.

Group			High-Fat Diet				
		Naive	Control	EFAX 25	EFAX 50	EFAX 100	Vitamin E
**Adipose tissue weight**	Epididymal (g)	0.79 ± 0.30	3.16 ± 0.13 ^##^	2.45 ± 0.40 *	2.60 ± 0.43	2.47 ± 0.35 *	2.99 ± 0.51
	Retroperitoneal (g)	0.26 ± 0.10	1.40 ± 0.10 ^##^	1.10 ± 0.21	1.12 ± 0.23	0.96 ± 0.18 *	1.31 ± 0.33
	Visceral (g)	0.54 ± 0.09	1.20 ± 0.10 ^##^	0.85 ± 0.17 *	0.92 ± 0.18	0.85 ± 0.12 *	1.22 ± 0.34
	Total (g)	1.60 ± 0.40	5.76 ± 0.28 ^##^	4.40 ± 0.73 *	4.64 ± 0.82	4.29 ± 0.60 *	5.52 ± 1.15
Total cholesterol (mg/dL)		89.9 ± 9.1	184.1 ± 15.3 ^##^	157.4 ± 4.8 **	138.7 ± 14.8 **	156.1 ± 8.5 **	183.6 ± 9.9
LDL-C (mg/dL)		7.9 ± 2.8	22.1 ± 1.6 ^##^	16.3 ± 0.9 **	15.2 ± 2.4 **	16.0 ± 1.6 **	23.0 ± 2.9
HDL-C (mg/dL)		90.0 ± 6.41	126.4 ± 8.7 ^##^	124.3 ± 7.2	118.5 ± 9.5	117.3 ± 9.1	126.6 ± 8.2
Triglyceride (mg/dL)		80.0 ± 15.0	44.2 ±14.9 ^##^	37.7 ± 3.4	36.2 ± 6.3	40.9 ± 6.5	45.3 ± 11.4

Data are expressed as mean ± standard deviation (SD). ^##^
*p* < 0.01 compared with naive group. * *p* < 0.05 and ** *p* < 0.01 compared with control group.

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
