# Peer review of "Ethyl Acetate Fraction of Amomum xanthioides Ameliorates Nonalcoholic Fatty Liver Disease in a High-Fat Diet Mouse Model"

_nutrients, 2020, doi:10.3390/nu12082433_

Round 1

Reviewer 1 Report

28th July 2020

Manuscript ID: Nutrients-890168

Ethyl Acetate Fraction of Amomum xanthioides Ameliorates Nonalcoholic Fatty Liver Disease in a High-Fat Diet Mouse Model

Comments to the Authors:

The purpose of the authors was to evaluate the anti-fatty liver effect of the ethyl acetate fraction of Amomum xanthioides (EFAX) on steatosis or non-alcoholic steatohepatitis, using a high-fat diet mouse model. The main results obtained were that EFAX treatment significantly ameliorated the steatohepatic changes, the increased weight of adipose tissues, and the altered serum lipid profiles. These observed effects were possibly due to the lipolysis-dominant activity of EFAX on multiple hepatic proteins including sterol regulatory element-binding protein (mSREBP)-1c, peroxisome proliferator-activated receptor (PPAR)-α, AMP-activated protein kinase, and diglyceride acyltransferases (DGATs).

This is an interesting study, however some major concerns have to be answered.

Major points:

  1. The products used to obtain EFAX could be toxic. How were calculated the quantities necessary to carry out the extraction in order to be safe?
  2. Are the authors sure that EFAX has not exerted any hyporexigenic effect?
  3. The authors give more importance to the effects on lipid metabolism-related proteins than on the inflammatory process or oxidative stress. Could they justify it?
  4. The effects of treatment could be not only due to changes in lipid metabolism but also to changes in intestinal dysbiosis present in NAFLD. It should be interesting to be mentioned in the Discussion section.

Minor points:

  1. Some sentences should be corrected: “Effects on in hepatic lipid profiles” (line 181). Revise the manuscript.
  2. There are some spelling errors: “Naive” instead of “Naïve”, “faction” instead of “fraction”. Revise it.

Author Response

Reviewer #1:

Major points:

  1. The products used to obtain EFAX could be toxic. How were calculated the quantities necessary to carry out the extraction in order to be safe?

â–º Thank reviewer for the meaningful comments and question. We decided the treated-concentration of extracts based on our previous animal studies (Lee SB 2016, Kim HG 2015). In the previous studies, EFAX 50 mg/kg did not show liver-toxicity according to serum levels of AST and ALT. In addition, before performing the main experiment of the present study, we conducted a pilot animal experiment treating EFAX 25 mg/kg, 50 mg/kg, and 100 mg/kg, and found no toxicity according to both survival rate and serum levels of AST and ALT (data not shown).

* Ref 1) Lee, SB et al. Ethyl Acetate Fraction of Amomum xanthioides Exerts Antihepatofibrotic Actions via the Regulation of Fibrogenic Cytokines in a Dimethylnitrosamine-Induced Rat Model. Evid.-Based Complement. Altern. Med. ECAM 2016.

* Ref 2) Kim, HG et al. Ethyl acetate fraction of Amomum xanthioides improves bile duct ligation-induced liver fibrosis of rat model via modulation of pro-fibrogenic cytokines. Sci. Rep. 2015.

  1. Are the authors sure that EFAX has not exerted any hyporexigenic effect?

â–º We absolutely agree with the reviewer’s concern. Thus, we measured the amount of food intake twice a week and confirmed that EFAX did not affect the amount of food intake, which means EFAX did not have hyporexigenic effects. We explained this in 160-163 lines, and provided the regarding data in Figure 1C,D.

  1. The authors give more importance to the effects on lipid metabolism-related proteins than on the inflammatory process or oxidative stress. Could they justify it?

â–º Thank reviewer for the constructive comment. Because fatty acid has been considered a key initiator for progression of NAFLD, we focused on lipid metabolism in the present study. However, as reviewer pointed out, NAFLD is multi-factorial disease involving numerous pathogenesis including inflammatory process and oxidative stress. It is not certain whether inflammation is always preceded by fat accumulation; however, lipid-overload liver injury is a dominant model for NASH pathogenesis. Therefore, we additionally explained this issue in the Discussion section.

  1. The effects of treatment could be not only due to changes in lipid metabolism but also to changes in intestinal dysbiosis present in NAFLD. It should be interesting to be mentioned in the Discussion section.

â–º We appreciate reviewer for the professional comments. As reviewer mentioned, currently intestinal dysbiosis is considered a critical contributor to metabolic disorders such as obesity and NAFLD. we added the description of the effects of intestinal dysbiosis on NAFLD in the Discussion section.

Minor points:

  1. Some sentences should be corrected: “Effects on in hepatic lipid profiles” (line 181). Revise the manuscript.

â–º Thank reviewer for the detailed correction. We removed “in” from the sentence (line 181).

  1. There are some spelling errors: “Naive” instead of “Naïve”, “faction” instead of “fraction”. Revise it.

â–º We appreciate reviewer for the correction. As reviewer pointed out, we corrected “Naïve” to “Naïve”. However, we did not understand the reason reviewer suggested to change “fraction” to “faction”. We suppose “fraction” is the right spelling in this manuscript.

Reviewer 2 Report

July 23, 2020                                                                                                                                                                                        nutrients
Manuscript ID: nutrients-890168
“Ethyl Acetate Fraction of Amomum xanthioides Ameliorates Nonalcoholic
Fatty Liver Disease in a High-Fat Diet Mouse Model”

In this manuscript, the authors reported that ethyl acetate fraction of Amomun xanthioides ameliorated NAFLD in a high-fat diet mouth model due to the modulation of molecules related to hepatic lipolysis, lipogenesis and secretion of lipids. This study is well-designed and the manuscript is well-described, but some revisions seem to be needed as stated in the comments to the authors.

Major comments:

1. Because this fraction was reported to possess an anti-hepatic fibrosis effect, I cannot help thinking why the authors did not determine its anti-hepatic fibrosis effect using NASH model with advanced fibrosis. Please explain this issue.

2. Lines 149-150: I am curious about dose-dependent effects of this fraction for NASH. Dunnett’s test may not be suitable for the evaluation of this dose-dependent effects.

3. The positive control group receiving vitamin E did not reveal the meaningful effects for NASH in most tests in this study. Please discuss the reasons for this. Due to dose? experimental periods?

4. Figure 4 (A): These panels should be needed for more detailed explanation, i.e. hepatic inflammation, hepatocyte ballooning and hepatic fibrosis histopathologically.

Minor comments:

1. Line 93-94: As the authors mentioned, “a 60% high fat diet” can be amended to “a 60% high fat and 20% low carbohydrate diet”.

2. Line 97: Please describe the euthanasia methods.

3. Figure 4: Semi-quantified graph B is missing.

Author Response

Reviewers’ comments:

Reviewer #2:

Major comments:

  1. Because this fraction was reported to possess an anti-hepatic fibrosis effect, I cannot help thinking why the authors did not determine its anti-hepatic fibrosis effect using NASH model with advanced fibrosis. Please explain this issue.

â–º Thank reviewer for the professional comments. As reviewer mentioned, our previous studies reported anti-fibrosis effects of EFAX. Because fatty acid has been considered a key initiator for progression of NAFLD, we decided to firstly investigate the effect of EFAX on NAFL (fatty liver) using a 8-week high fat diet model. Our next study will certainly investigate the effect of EFAX on NASH with advanced fibrosis using a 16-week high fat model.

  1. Lines 149-150: I am curious about dose-dependent effects of this fraction for NASH. Dunnett’s test may not be suitable for the evaluation of this dose-dependent effects.

â–º We absolutely agree with the reviewer’s comment. As reviewer pointed out, we further performed post-hoc analysis with Turkey’s test, which can compare each experimental group. However, we did not find any dose-dependent effect of EFAX in the present study. As well-known, Dunnett’s test is a common method to compare control group and experimental groups.

  1. The positive control group receiving vitamin E did not reveal the meaningful effects for NASH in most tests in this study. Please discuss the reasons for this. Due to dose? experimental periods?

â–º We fully agree with the reviewer’s opinion. We cannot assure the reasons for the non-positive effects of vitamin E in the present model. As we briefly described in the Discussion section, previous clinical studies also reported controversial results. In addition, previous animal studies using high fat diet also reported non-positive results with vitamin E 100 mg/kg (Sakr, H.F 2018, ÅžekeroÄŸlu, V. 2018). We supplemented the description of this issue in the Discussion section.

* Ref 1) Sakr, H.F et al. Swimming, but not vitamin E, ameliorates prothrombotic state and hypofibrinolysis in a rat model of nonalcoholic fatty liver disease. J. Basic Clin. Physiol. Pharmacol. 2018.

* Ref 2) ÅžekeroÄŸlu, V. et al. Hepatoprotective effects of capsaicin and alpha-tocopherol on mitochondrial function in mice fed a high-fat diet. Biomed. Pharmacother. Biomedecine Pharmacother. 2018.

  1. Figure 4 (A): These panels should be needed for more detailed explanation, i.e. hepatic inflammation, hepatocyte ballooning and hepatic fibrosis histopathologically.

â–º Thank reviewer for the helpful comment. As reviewer suggested, we added the detailed explanation on Figure 4A including hepatic inflammation, hepatocyte ballooning and hepatic fibrosis in the Result section, and marked ballooned hepatocytes in Figure 4A.

Minor comments:

  1. Line 93-94: As the authors mentioned, “a 60% high fat diet” can be amended to “a 60% high fat and 20% low carbohydrate diet”.

â–º Thank reviewer for the helpful comment. As reviewer suggested, we amended “a 60% high fat diet” to “a 60% high fat and 20% low carbohydrate diet” (line 93-94).

  1. Line 97: Please describe the euthanasia methods.

â–º Thank reviewer for the detailed suggestion. We added the description on the euthanasia method in the material and method section (line 97-99).

  1. Figure 4: Semi-quantified graph B is missing.

â–º Thank reviewer for finding out a critical error. We added a semi-quantified graph on H&E staining (Figure 4B).

Round 2

Reviewer 1 Report

The authors have adequately answered all the questions and have improved the text, so I consider the manuscript suitable for publication in present form.